An unusual early-diverging plesiosauroid from the Lower Jurassic Posidonia Shale of Holzmaden, Germany

Sachs Sven sachs.pal@gmail.com 1
Madzia Daniel 2
1 Abteilung Geowissenschaften, Naturkunde-Museum Bielefeld , Bielefeld , Germany
2 Department of Evolutionary Paleobiology, Institute of Paleobiology, Polish Academy of Sciences , Warsaw , Poland
.Zyła Dagmara
Electronic publication date: 2025 Aug 4
Publication date: 2025
Volume: 13
Electronic Location ID: e19665
Received 2024 Aug 23; Accepted 2025 Jun 5
Copyright: ©2025 Sachs and Madzia
Copyright year: 2025
Copyright holder: Sachs and Madzia
License: This is an open access article distributed under the terms of the Creative Commons Attribution License, which permits unrestricted use, distribution, reproduction and adaptation in any medium and for any purpose provided that it is properly attributed. For attribution, the original author(s), title, publication source (PeerJ) and either DOI or URL of the article must be cited.
License URL: https://creativecommons.org/licenses/by/4.0/

Keywords: Plesiosauroidea, Plesiosauria, Toarcian, Jurassic, Osteology, Phylogeny reconstruction, Holzmaden, Germany

Funding: National Science Centre, Poland 2023/51/B/NZ8/00899 This research was funded by National Science Centre, Poland, grant no. 2023/51/B/NZ8/00899. The funders had no role in study design, data collection and analysis, decision to publish, or preparation of the manuscript.

==============================
The lower Toarcian Posidonia Shale at Holzmaden, Southwest Germany, has yielded some of the most remarkable Lower Jurassic marine tetrapod specimens, including five plesiosaur taxa identified from nearly complete skeletons. This study provides a comprehensive description of an osteologically immature plesiosauroid skeleton found in a Holzmaden quarry in 1978. Despite that the specimen has been researched in the past, previous studies have been either brief or targeted some specific aspects of the specimen, such as its soft tissue preservation. The anatomy and taxonomy of the specimen have never been explored in detail. We reinterpret several of its osteological features and evaluate their taxonomic and phylogenetic significance. Our findings reveal that the specimen possesses an unusual combination of character states, which are not markedly affected by ontogenetic development, warranting the designation of a new taxon, Plesionectes longicollum gen. et sp. nov., thereby increasing the known plesiosaur diversity of both the Toarcian age and the Posidonia Shale.

Introduction

The lower Toarcian (upper Lower Jurassic) Posidonia Shale (Posidonienschiefer Formation) is widely recognized for its record of exquisitely preserved fossils (e.g., Hauff, 1921; Hauff, 1953; Hauff & Hauff, 1981). The most spectacular of them were unearthed from the Holzmaden area, lying approximately 35 km east of Stuttgart in Southwest Germany. To date, five plesiosaur taxa have been described from the Posidonia Shale at Holzmaden, each known from almost complete skeletons. These include the early-diverging pliosaurid Hauffiosaurus zanoni (O’Keefe, 2001; Vincent, 2011), the rhomaleosaurid Meyerasaurus victor (Fraas, 1910; Smith & Vincent, 2010), and the plesiosauroids Seeleyosaurus guilelmiimperatoris (Dames, 1895; Fraas, 1910; Sachs et al., 2025), Microcleidus brachypterygius (Von Huene, 1923; Großmann, 2006; Großmann, 2007), and Plesiopterys wildi (O’Keefe, 2004; Marx et al., 2025). Other significant plesiosaur specimens originating from the Posidonia Shale include a nearly complete, peculiar skeleton pertaining to an early juvenile individual with uncertain taxonomic affinities (Vincent, 2010), and several unusual cervical vertebrae from Bavaria, described under the name ‘Plesiosaurus’ bavaricus (Dames, 1895; Sachs, Abel & Madzia, 2023). These specimens demonstrate that the diversity of plesiosaurs from the Posidonia Shale is remarkably high and still incompletely understood.

In 1978, a nearly complete and largely articulated skeleton of a plesiosauroid has been discovered by Gotthilf Fischer in a quarry in Holzmaden. The specimen was prepared by its discoverer and was obtained by the Staatliches Museum für Naturkunde in Stuttgart (SMNS) in 1979 where it has since been curated under the catalog number SMNS 51945.

The specimen has previously been figured by Urlichs, Wild & Ziegler (1994, fig. 18) and Ziegler (1986, fig. 117, 1994, fig. 50) and assigned to Plesiosaurus. Wild (1989, fig. 86) treated it as a probable representative of Microcleidus, while Großmann (2006), who provided a brief description of the specimen, noted that it probably belongs to a novel species. More recently, SMNS 51945 was evaluated by Vincent et al. (2017); however, this study mainly focused on its soft tissue. Although Vincent et al. (2017: p. 8) also concluded that SMNS 51945 “differs substantially from all the lower Toarcian plesiosauroids previously described [and] from coeval plesiosauroid specimens known from the same region”, indicating that it likely represents a taxon not yet known from the Posidonia Shale, they refused to introduce a new name for the taxon due to the ontogenetic stage of the specimen and poor preservation of its skull, and considered SMNS 51945 as an indeterminable plesiosauroid.

Here, we provide the first detailed osteological description of SMNS 51945, reinterpreting some of the osteological features identified by Großmann (2006) and Vincent et al. (2017), and assess its taxonomic significance and phylogenetic affinities. We demonstrate that SMNS 51945 bears an unusual combination of character states whose appearance is not markedly affected by ontogenetic development.

The presence of a unique suit of osteological characters found in SMNS 51945 justifies the erection of a new genus and species, which adds to the known plesiosaur diversity of both the Toarcian age and the Posidonia Shale.

Methods

Protocol for phylogenetic analyses

We explored the phylogenetic affinities of SMNS 51945 using the matrix obtained from Sachs et al. (2025), which represents a significantly modified version of the dataset originally assembled by Benson & Druckenmiller (2014). Except for adding SMNS 51945, only a single additional change has been made: replacement of Plesiopterys wildi with first-hand scores of SMNS 16812 (holotype of P. wildi) and MH 7 (referred specimen of P. wildi), following Marx et al. (2025). The final version of the matrix comprised 132 operational taxonomic units (OTUs) scored for 270 characters; 67 of which were set as ‘additive’ (= ‘ordered’), as proposed by Madzia, Sachs & Lindgren (2019).

We used maximum parsimony as the optimality criterion and performed our analyses in TNT 1.6 (Goloboff & Morales, 2023). We have conducted three runs. The first run was based on equal weights and the additional two runs used the implied weighting function, setting the concavity parameter (K) to 9 and 12. In all analyses we used Neusticosaurus pusillus as the outgroup. Each of the analyses has been performed using the same settings. We first fixed the maximum number of most parsimonious trees to 200,000 trees by adding the command “hold 200000” directly into the TNT script. Upon opening it in TNT, we ran the ‘New Technology’ search, involving 500 addition sequences and default settings activated for sectorial searches, ratchet, drift, and tree fusing. After obtaining the first results, we performed an additional ‘Traditional’ search with tree bisection-reconnection (TBR) branch-swapping using trees saved to RAM. For the analysis using equal weighting, we calculated the Bremer support using TBR and retaining sub-optimal trees with up to three additional steps. Nodal support for the analyses with implied weighting was determined using Symmetric Resampling; a ‘Traditional’ search was performed with 1,000 replicates, a default change probability set at 33, and output expressed as frequency differences (GC).

See Supplemental Information 1 for the character list and Supplemental Information 2 for a TNT-executable file.

Nomenclatural acts

This publication, along with the nomenclatural acts it includes, has been formally registered with ZooBank, the official online registry of the International Code of Zoological Nomenclature (ICZN). Each act is associated with a unique Life Science Identifier (LSID), which can be resolved and accessed via any standard web browser by appending the LSID to the URL prefix https://zoobank.org/. The LSIDs corresponding to this work are as follows: urn:lsid:zoobank.org:pub:23DBDB92-0752-4068-ACC5-A29D217D6CE3 for the publication itself; urn:lsid:zoobank.org:act:C1A1E982-0895-46F5-9B21-F9741E111949 for the newly described genus Plesionectes; and urn:lsid:zoobank.org:act:73D43879-D9DA-410A-8F2D-F48BC1033432 for the new species Plesionectes longicollum.

Systematic paleontology

Plesiosauria De Blainville, 1835.	
Plesiosauroidea Gray, 1825.	
Plesionectes gen. nov.	

Type species. Plesionectes longicollum sp. nov.

Etymology. The name Plesionectes derives from plēsíon (Greek), meaning “close”, “near”, referring to its plesiosaur affinities, and nēktēs (Greek), “swimmer”, common suffix in plesiosaur taxon names.

Diagnosis. As for the type and only species.

Plesionectes longicollum sp. nov.

Type specimen. SMNS 51945 (Fig. 1), a nearly complete, largely articulated skeleton, comprising a partially exposed skull including the exoccipital-opisthotic, quadrates, and part of the squamosals, 41 articulated cervical vertebrae, four pectoral vertebrae, 20–21 dorsal vertebrae, two sacral vertebrae and 39 caudal vertebrae and associate ribs, gastralia, incomplete pectoral girdle with interclavicle, left scapula and part of right scapula, fragmentary coracoids, incomplete pelvic girdle with complete left ilium, part of right ilium, both ischia, incomplete pubes, complete left front and hind limb, disarticulated distal right front limb, and right hind limb.

Figure 1 Skeleton of Plesionectes longicollum gen. et sp. nov. (SMNS 51945).

Scale bar equals 30 cm.

Etymology. The name longicollum derives from longus (Latin), meaning “long”, and collum (Latin), “neck”, in reference to its long neck, comprising at least 43 vertebrae.

Type locality and horizon. Holzmaden, Esslingen District, Baden-Württemberg, southwestern Germany; lower Toarcian (Lias ɛII1, ‘Koblenzer’, Dactylioceras tenuicostatum Zone, D. semicelatum Subzone; Hauff, 1921; Riegraf, Werner & Lörcher, 1984; Maisch, 2021), Lower Jurassic.

Diagnosis. Plesiosauroid plesiosaur bearing following unique combination of characters states: paraoccipital process being considerably longer as the height of the exoccipital body; neck comprising ≥43 cervical vertebrae; V-shaped neurocentral suture in the cervical and pectoral vertebrae (potential local autapomorphy sensu Beeston et al., 2024); conjoined parapophysis and diapophysis in the anterior, middle, and the majority of the posterior cervicals, one rib facet formed in the posteriormost cervical vertebrae; cervical rib processes strap-shaped and pronounced in anterior and mid-neck region; posterior cervical and pectoral neural spines not considerably taller than long (mostly < 1:2), lacking constriction at base; dorsal vertebral series comprising 20–21 vertebrae.

Description and comparisons

Measurements

The length of the skeleton, as preserved, is 295 cm. The neck is 125 cm long; the pectoral and dorsal series is 85 cm long; and the tail is 81 cm long. The estimated length of the individual, including the skull, is 320 cm. Detailed measurements of the elements pertaining to the skeleton of SMNS 51945 are provided in Tables S1–S3.

Skull

General remarks. The skull is heavily damaged and only a few elements can be identified (Fig. 2). Vincent et al. (2017) described the preserved skull elements to include a dorsally misplaced mandibular ramus, the exoccipital-opisthotic, and a portion of the squamosal (Vincent et al., 2017, fig. 2a). These cranial elements were not included in their description of the specimen. We agree with the identification of the exoccipital-opisthotic and ventral cranial elements as part of the squamosal. However, we consider both laterally placed elements as the articulated quadrates and squamosals that were pushed anteriorly when the skull collapsed and are now accessible in posterior view. The remaining parts of the skull cannot be identified with certainty due to poor preservation. For the same reason, not much can be measured. The paraoccipital process is 22.6 mm long; the exoccipital is 15 mm high and 10.4 mm long (ventrally).

Figure 2 Exposed skull elements of Plesionectes longicollum gen. et sp. nov. (SMNS 51945).

(A) Overview. (B) Close-up of the exoccipital-opisthotic. (C) Close-up of the right squamosal and quadrate. Scale bars equal three cm in A and one cm in B and C. Abbreviations: cau, chamber for the ampulla and utriculus; exo, exoccipital-opisthotic; fcv, first preserved cervical vertebra; for, foramina; pop, paraoccipital process; qu, quadrate; sq, squamosal; X–XI, cranial nerve foramina, probably for vagus (X) and accessory (XI) nerves.

Squamosal. Both squamosals are exposing their posterior aspect and they are still in articulation with the quadrates (Fig. 2A). From the right squamosal, the posterior side, along with portions of the lateral and medial aspects, is exposed. Most of the squamosal arch is likewise visible. The left squamosal is exposed from the medial side of the arch. Both squamosals indicate the initial curvature of the arch, though the right one may be somewhat distorted. The transverse width and anteroposterior length of the squamosal arch appear to be equal. The squamosal apex and the anterior ramus of the squamosal are not exposed.

Quadrates. Both quadrates are preserved in articulation with the squamosals (Fig. 2A). The right quadrate is exposed from its posterior aspect; the left one is accessible from the medial side. Both quadrates are high-rectangular elements and appear to have a pointed dorsal side. The quadrate condyles are damaged.

Exoccipital-opisthotic. The complete right exoccipital-opisthotic is exposed in anterolateral view (Fig. 2). The dorsal side of the exoccipital appears straight but is partly obscured by a cervical centrum. Anterodorsally, there is a concave edge, which might have been part of the chamber for the ampulla and utriculus (compare Benson, Evans & Taylor, 2015, fig. 8F) (Fig. 2B). Ventral to this concave edge, there is a slight anterior protrusion ventral to which the anteroventral side of the exoccipital appears straight. On the ventral side of the exoccipital-opisthotic there are remnants of one or two hypoglossal foramina (XII). Laterally, a prominent, ventrally inclined paraoccipital process is formed. The paraoccipital process, which is considerably longer than the height of the body of the exoccipital (Fig. 2B), is gradually dorsoventrally expanded at its posterolateral end. The latter is partly damaged, but it seems that the process is otherwise complete. A prominent oval foramen for vagus (X) and accessory (XI) nerves is present directly ventral to the paraoccipital process. The posterior side of the exoccipital-opisthotic is not exposed.

Axial skeleton

Post-axis cervical vertebrae and ribs. The neck is completely articulated (Fig. 1), but in particular cervicals 8–10, 26–28, and 32–39 bear brownish repair plaster that was added during the preparation. These vertebrae are still largely genuine. The specimen bears 41 post-axis cervical vertebrae. The first preserved vertebra can be identified as a post-axial cervical as its matches the morphology and the dimensions of the other anterior cervical vertebrae and because it lacks characters typically found in the atlas-axis complex of other Toarcian plesiosauroids such as a hypophyseal ridge or single axis rib facet (compare e.g., Seeleyosaurus guilelmiimperatoris (Sachs et al., 2025)).

Großmann (2006) mentioned that 35 cervicals are preserved and Vincent et al. (2017) stated that 42 to 43 cervicals, including the atlas-axis, are the minimum cervical count. We estimate that the complete neck comprised at least 43 vertebrae.

This number is considerably higher than in most other Early Jurassic plesiosauroids (see Table 1). The seven anteriormost and four posteriormost cervicals are exposed in ventral view, the remaining cervicals are shown in lateroventral view. All cervicals are still articulated with one another, thus the morphology of the articular faces of the centra cannot be described. Only the sharp lateral rims of the articular faces are visible.

Table 1 Selected characters compared in early-diverging plesiosauroids.

	Plesionectes longicollum	Westphaliasaurus simonsensii	Seeleyosaurus guilelmiimperatoris	Microcleidus tournemirensis	Microcleidus brachypterygius	Microcleidus homalospondylus	Microcleidus melusinae	Plesiopterys wildi	‘ Plesiosaurus ’ bavaricus	Franconiasaurus brevispinus	
Number of cervical vertebrae	≥43	<30	35–36	43	36	40	32	39	Unknown	<30	
Rib facets of anterior and middle cervical vertebrae	Two co-joined rib facets	Two co-joined rib facets	Two co-joined rib facets	Two co-joined rib facets	Two co-joined rib facets	Two co-joined rib facets	Two co-joined rib facets	Two co-joined rib facets	One rib facet	Two co-joined rib facets	
Rib facets of posterior cervical vertebrae	Mix of two co-joined and one rib facet	One rib facet	Two co-joined rib facets	Two broadly separated rib facets	Two co-joined rib facets	Two broadly separated rib facets	Two co-joined rib facets	Two co-joined rib facets	One rib facet	Two co-joined rib facets	
Rib processes in anterior cervical vertebrae	Marked anterior and posterior processes	Reduced anterior processes	Marked anterior and posterior processes	Marked anterior and posterior processes	Marked anterior and posterior processes	Marked anterior and posterior processes	Marked anterior and posterior processes	Reduced anterior processes	Unknown	Reduced anterior processes	
Mid-ventral surface of cervical centra	Flat	Flat	Flat	Flat	Unknown	Flat	Flat	Flat	Sharp midline ridge	Some flat, some bear rounded midline ridge	
Shape of neurocentral suture in anterior and middle cervical vertebrae	V-shaped	Evenly convex, extends far ventrally	Unknown	Rounded	Rounded	Rounded	Unknown	Rounded	V-shaped	Unknown	
Number of pectoral vertebrae	4	2	5	3	5	5	2–4	4	Unknown	2	
Number of dorsal vertebrae	20–21	20–22	15–17	16	14	17	Unknown	19	Unknown	22	
Height of dorsal neural spines	Less than twice as high as the centrum	Less than twice as high as the centrum	More than twice as tall as the centrum	More than twice as tall as the centrum	More than twice as tall as the centrum	More than twice as tall as the centrum	Unknown	Less than twice as high as the centrum	Unknown	Less than twice as high as the centrum	
Base of dorsal neural spines	Unconstricted	Unconstricted	Contricted	Contricted	Contricted	Contricted	Contricted	Unconstricted	Unknown	Unconstricted	
Caudal ribs facet location in proximal and middle caudal vertebrae	Located far dorsally	Located far dorsally	Located dorsally	Unknown	Located dorsally	Located dorsally	Unknown	Located far dorsally	Unknown	Located far dorsally	
Inclination of proximal humerus end	Not inclined	Not inclined	Inclined posteriorly	Inclined posteriorly	Inclined posteriorly	Inclined posteriorly	Inclined posteriorly	Not inclined	Unknown	Not inclined	
Proportions of hind limb epipodials	Fibula larger	Size about equal	Fibula larger	Fibula larger	Size about equal	Fibula larger	Unknown	Fibula larger	Unknown	Tibia larger	

The lateral sides of the centra are only partly exposed, but a longitudinal lateral ridge (Fig. 3A) is indicated in the vertebrae 12 to 23 of the preserved cervical series. Vincent et al. (2017) stated that the centra do not possess a lateral longitudinal ridge. The preservation and distortion of the centra complicates the identification of the lateral aspects of the centra, but the lateral ridge that we trace is found at about the same position in all of the centra. Among Toarcian plesiosauroids a lateral longitudinal ridge was described for Microcleidus tournemirensis, M. homalospondylus, and Seeleyosaurus guilelmiimperatoris (Bardet, Godefroit & Sciau, 1999; Owen, 1865; Fraas, 1910).

Figure 3 Cervical vertebrae of Plesionectes longicollum gen. et sp. nov. (SMNS 51945).

(A) Anterior cervicals in ventrolateral view. (B) Anterior cervical ribs in medial view. (C) Anterior cervicals in ventral view. (D) Posterior cervicals in lateral view. Scale bars equal three cm. Abbreviations: ans, anterior cervical neural spine; ap, anterior cervical rib process; dps, diapophysis; fs, subcentral foramen; llr, longitudinal lateral ridge; ncs, neurocentral suture; pns, posterior cervical neural spine; poz, postzygapophyses; pp, posterior cervical rib process; pps, parapophysis; prz, prezygapophyses; srf, single cervical rib facet.

The largely complete neurapophyses are unfused, disarticulated from the centra and exposed in lateral view. A broad V-shaped, thus ventromedially pointed, neurocentral suture which does not extend far ventrally, is evident in nearly all cervicals (Figs. 3A, 3D). The horizontally facing zygapophyses are still in articulation with one another and the zygapophyseal facets, which appear planar, are only visible in few vertebrae.

The neural spines are high-rectangular in lateral view and appear to have been only slightly higher than the accompanying centra (Figs. 3A, 3D). A constricted base, typically found in microcleidids (Benson, Evans & Druckenmiller, 2012), is not evident in SMNS 51945. The anterior neural spines are straight and flare anteroposteriorly at their dorsal end, thus having a slightly anteriorly directed anterodorsal side, and a posterodorsally directed posterior side (Fig. 3A). In the mid-neck level the neural spines start to curve slightly posteriorly, and in the posteriormost cervicals the neural spines bear a gently convex anterior and a distinctly concave posterior edge. The dorsal aspects of the neural spines are gently convex in lateral view throughout the neck and do not appear to be transversely expanded. The anterior cervical neural spines are slightly longer at the base than they are high (ratio 0.80 in cervical 12), in the mid-neck the neural spines are slightly higher than long (ratio 1.25 in cervical 24), and they are clearly higher than long in the posteriormost cervicals (ratio 1.96 in cervical 40). In contemporary microcleidids the posterior cervicals neural spines reached a height of more than three times the length (Bardet, Godefroit & Sciau, 1999; Sachs et al., 2025). Even though in plesiosaurs neural spine proportions could change during ontogeny, we consider it unlikely that the neural spines in Plesionectes longicollum gen. et sp. nov. became significantly higher and we consider it more likely that the taxon had lower neural spines comparable with those in Franconiasaurus brevispinus (Sachs, Eggmaier & Madzia, 2024).

The ventral sides of the centra are either gently concave or flat and bear a pair of small foramina subcentralia (Fig. 3C), a condition commonly found in Early Jurassic plesiosauroids (Benson & Druckenmiller, 2014, appendix 2, characters 156 and 165). As noticed by Vincent et al. (2017), there is no pronounced midline keel separating the foramina. Instead, the area is either flat or gently concave and slightly rounded only in the mid-section of the neck. This differs from the condition in ‘Plesiosaurus’ bavaricus, where a sharp midline keel is found (see Sachs, Abel & Madzia, 2023). The lateroventrally placed rib facets are exposed in most centra. A conjoined diapophyseal and parapophyseal facet is developed in the majority of the cervicals (Fig. 3B). However, in cervical 37, a shift takes place and the posterior five cervicals are single headed (Fig. 3D). This condition is shared with Westphaliasaurus simonsensii (Schwermann & Sander 2011) while in the posterior cervical vertebrae of other Pliensbachian and Toarcian plesiosauroids two rib facets are formed (Table 1). Two conjoined rib facets are also found in the posteriormost cervical vertebra of the Sinemurian plesiosauroid Plesiopharos moelensis (Puértolas-Pascual et al., 2021). The diapophysis and parapophysis are similar in size and both are long-oval to slightly triangular. The cervical ribs bear distinct anterior and posterior processes. The anterior processes are more triangular in shape, being shorter and dorsoventrally higher than the posterior ones that are more rod-shaped (Fig. 3B). In the posterior cervical region, where the cervical ribs become single-headed, the transition to the pectoral and dorsal ribs takes place, from the typical cervical rib shape with two distinct processes to reduced anterior and elongated posterior processes (Fig. 3D).

Pectoral vertebrae and ribs. An articulated series of pectoral vertebrae is preserved (Fig. 3), but the exact number of vertebrae is difficult to infer. Großmann (2006) stated that five pectorals are present, while Vincent et al. (2017) suggested 3–4 pectorals. However, the rib facet at the centrum is only visible in the first vertebra of the series. In the other pectorals it is obscured by overlying ribs or neurapophyses. We consider the 42nd vertebra of the series to be the first pectoral, because part of the rib facet is evident on the neural arch and the rib facet that is visible on the centrum migrated dorsally, relative to the position in the 41st vertebra which then is the last cervical. In the second pectoral (vertebra 43) only part of the rib facet is visible on the centrum as it is largely obscured. On the neural arch of the 44th and 45th vertebra, a larger and more circular rib fact is evident. However, the heads of the accompanying ribs are clearly larger than the exposed rib facets. This indicates that part of the rib articulation took place on the centrum. Vertebra 46 of the series shows a more prominent and high-oval rib facet, placed on the neural arch. This appears to be the first dorsal vertebra. Hence, four pectoral vertebrae seem to be the actual count. Microcleidus brachypterygius (Großmann, 2007) and Seeleyosaurus guilelmiimperatoris (Sachs et al., 2025) both bear five pectoral vertebrae, whereas Plesiopterys wildi (O’Keefe, 2004) and Plesiopharos moelensis (Puértolas-Pascual et al., 2021) have four pectorals.

Figure 4 Post-cervical axial skeleton of Plesionectes longicollum gen. et sp. nov. (SMNS 51945).

(A) Cervicodorsal transition. (B) Anterior and mid-dorsal vertebrae and proximal ribs. (C) Distal dorsal ribs and gastralia. (D) Sacral and proximal caudal vertebrae. (E) Proximal caudal vertebrae. (F) Distal caudal vertebrae. Scale bars equal three cm. Abbreviations: b.na, base of neural arch pedicle; cdr, caudal rib; crf, caudal rib facet; dr, dorsal rib; fca, first caudal vertebra; fd, first dorsal vertebra; fp, first pectoral vertebra; ga, gastralia; hsf, hemapophyseal facet; lc, last cervical vertebra; nbf, facet for base of neural arch pedicle; ns, neural spine; rf, rib facet; sr, sacral rib; srf, sacral rib facet; st, soft tissue; sv, sacral vertebra; tp, transverse process.

The articular faces of the centra are not visible, but the exposed rims, surrounding the faces, are sharp. The pectoral centra have a flat ventral side and bear foramina subcentralia (Fig. 4A). The neurocentral suture in the pectorals is V-shaped, resembling the sutures observed in the cervicals, and the neural spines have a height to length ratio (measured at the base) of 1.94–2.11. Unlike the posterior cervical neural spines, those of the pectorals are straight and not posteriorly curved (Fig. 4A).

Ribs are preserved near the second, third, and fourth pectoral vertebra. They have been tilted anteromedially and display their lateral and part of their posterior aspects. The rib heads widen anteroposteriorly. The distal parts of the ribs are likewise widened anteroposteriorly and overlap each other.

Dorsal vertebrae and ribs. The dorsal vertebrae are largely obscured by matrix and overlying dorsal ribs (Figs. 1, 4B, 4C). The neurapophyses in the anterior dorsals (with the exception of the first one) are disarticulated from the centra and tilted to the side so that their ventral aspects are exposed (Fig. 4B). The accompanying centra are likewise visible in ventral view. In about the middle of the dorsal vertebral column the centra have moved further ventrally but it is unclear if they show the ventral or lateral aspect. At the same time the disarticulated neurapophyses of these vertebrae have been shifted dorsally. They are titled in the opposite way as the previous neurapophyses, thus showing the dorsal aspect of the transverse processes and the neural spines. Because of the named preservation it is difficult to infer the exact vertebral number.

Großmann (2006) stated that 23–24 dorsal vertebrae are present while Vincent et al. (2017) counted 17–19 dorsal vertebrae. We consider 20–21 vertebrae (number 46–66 in the series) as dorsals, in which it remains unclear if the last dorsal vertebra (number 66) is in fact the first sacral since the centrum is exposed only from its dorsal aspect. Seeleyosaurus guilelmiimperatoris preserves 15 dorsals in SMNS 12039 and 17 in MB.R.1992 (Sachs et al., 2025), Microcleidus brachypterygius has 14 dorsals (Großmann, 2007), and Plesiopterys wildi bears 19 dorsals (O’Keefe, 2004).

In the anteriormost centra there is a broadly rounded ventral midline keel, adjacent to which there is a pair of foramina subcentralia. The articular faces are partly exposed in some centra and are weakly concave. The articular surface rims are sharp as in the pectoral vertebrae (Fig. 4B). The facets of the neural arch pedicles have a wide triangular shape. The transverse processes are rather short (Fig. 4B) and it seems that their bases were located above the neural canal, though this cannot be said with certainty. A single slightly concave and long-oval rib facet is formed. The zygapophyses are only visible in some dorsal vertebrae and here the prezygapophyses are gently dished. The neural spines have a height to length ratio (measured at the base) between 1.8 and 1.67.

The dorsal ribs are placed at the accompanying centra (Figs. 1, 4B). As in the pectoral ribs they have been titled anteromedially and display their lateral and part of their posterior sides. The general morphology matches that of the pectoral ribs with the rib heads being anteroposteriorly expanded, and having a slightly convex facet. The distal parts of the ribs are also widened anteroposteriorly, being considerably wider than the ribs midshaft, but also wider as the distal ends in the pectoral ribs. They overlap each only partly.

Gastralia. The gastral basket is disarticulated but it seems that most gastralia are present, being placed ventral to the dorsal ribs (Fig. 4C). The gastral ribs largely overlap each other and some elements also show plaster used for repair. As in other plesiosaurs (e.g., Fraas, 1910; Brown, 1981), the gastralia are strap-shaped elements and most of them show only one pointed side which identifies them as lateral elements (see, e.g., Fraas, 1910, fig. 3). Near the hind limb there are several slightly boomerang-shaped elements. They are exposed in posterior view, have two pointed sides and represent central elements.

Sacral vertebrae and ribs. Two vertebrae, being placed just dorsal to the left ilium, can be identified as sacral vertebrae because of their prominent and dorsoventrally high rib facets (Fig. 4D). The next vertebra in the series can be identified as a caudal because of the more anteriorly placed and clearly smaller rib facet that is also seen in other proximal caudals. The identification of the vertebra in front of the first sacral is less clear as the centrum is tilted and shows the dorsal aspect only. It thus remains unclear whether this is the last dorsal or another sacral vertebra. Vincent et al. (2017) considered 4–5 sacrals to be present, while Großmann (2006) noticed the presence of two to three sacral vertebrae in SMNS 51945. Seeleyosaurus guilelmiimperatoris bears three sacral vertebrae (Sachs et al., 2025). Two sacral vertebrae have been reported for Microcleidus brachypterygius (Von Huene, 1923), three for Microcleidus homalospondylus (Watson, 1909) and Plesiopterys wildi (O’Keefe, 2004), and four for Microcleidus tournemirensis (Bardet, Godefroit & Sciau, 1999).

The sacral vertebrae are visible in lateral view with the ventral sides of the centra being obscured by the associate ribs (Fig. 4D). The ovoid rib facet in the first sacral is placed in the anteroposterior middle of the centrum and occupies most of its lateral side. The same seems to be true for the second sacral, which, however is slightly anteriorly tilted and thus partly obscured by the first sacral. In both vertebrae the rib facet is indented and surrounded by a sharp edge. Dorsally, the sacral rib facets are connected with the base of the neural arch and have been roofed by the pointed lateroventral side of the neural arch pedicle, which participated in the formation of the rib facet in both sacrals. The sacral neurocentral sutures are more rounded than V-shaped. The morphology of the zygapophyses resembles that of the dorsal vertebrae. The neural spines have a height to length ratio of approximately 1.7 and it seems that the dorsal side of each neural spine is slightly more anteroposteriorly widened than the ventral side.

Two disarticulated sacral ribs are present adjacent to the sacral centra (Fig. 4D). The rib shafts are mostly obscured by matrix or other ribs. The rib heads are slightly convex and match the size and ovoid shape of the corresponding rib facets.

Caudal vertebrae and ribs. The tail of SMNS 51945 comprises 39 vertebrae. The same number was reported by Großmann (2006), whereas Vincent et al. (2017) counted 40–41 caudals. The tail is present in articulation (Fig. 1); however, some vertebrae have been shifted. The six proximal most caudal vertebrae are present in lateral to laterodorsal view. The seventh caudal is tilted ventrally and exposes the dorsal side with the facets for the neural arch pedicles. In the following vertebrae the centra are exposed in ventral view. The 11th and 12th caudal vertebrae are visible in lateral view. Then the tail is once again tilted in a way that in the 13th caudal the ventral and part of the lateral side are exposed. Most of the following vertebrae are visible in ventral view. Only the posteriormost caudals may be twisted and appear to show the dorsal side. All distalmost caudals are preserved isolated and they do not form a pygostyle-like structure (Fig. 4F).

The proximal caudal vertebrae can be distinguished from the sacrals by their smaller rib facets which are placed near the anterolateral side of the centra (Fig. 4D). The rib facets are circular and indented. Two vertebrae in the mid-section of the tail are visible in lateral view and show that the rib facets have migrated more dorsally, but they are still placed anterolaterally, thus having a certain distance to the posterior edges of the centra (Fig. 4E). In these vertebrae, the dorsal side of the rib facet is open and was thus formed by the neural arch pedicle. In the proximal caudals the neurapophyses are still articulated with the centra and the neurocentral suture seems to be rounded (Figs. 4D, 4E). The prezygapophyses appear to exceed the centrum anteriorly with most of their length, while the postzygapophyses only exceed the centra with about half of their length. The neural spines of the proximal caudals have a height to length ratio of about 1.4 (the ratio in the distal caudals cannot be measured). The shape of the neurapophysis resembles that of the dorsal vertebrae.

The articular surfaces of the caudal centra cannot be seen, but the articular surface rims are sharp as in the other vertebrae. Hemapophyseal facets are visible in the vertebrae that are preserved in ventral view. They are present on the posterior side of the centra being placed posterolaterally. Some rectangular caudal ribs and strap-shaped hemapophyses are present adjacent to the proximal caudal vertebrae and partly overlap each other (Fig. 4E).

Appendicular skeleton

Interclavicle. The interclavicle is a T-shaped element that is placed anterior to the scapula (Fig. 5A). The interclavicle is preserved in ventral aspect, showing the semi-oval and wing-shaped lateral sides and the rod-shaped posterior side. Anteromedially, the wing-shaped parts of the interclavicle are incised by a deep notch. The overall morphology differs from the interclavicle in Seeleyosaurus guilelmiimperatoris (Fraas, 1910; Sachs et al., 2025), Microcleidus tournemirensis (Bardet, Godefroit & Sciau, 1999), Plesiopterys wildi (O’Keefe, 2004) and Westphaliasaurus simonsensii (Schwermann & Sander 2011) where the anteromedial aspect is widely concave. A similar deep notch was reconstructed for Plesiosaurus dolichodeirus (Storrs, 1999, fig. 10). The posterior side of the interclavicle of SMNS 51945 is broken and partly shifted but a rod-shape morphology, similar to that in Microcleidus tournemirensis (Bardet, Godefroit & Sciau, 1999), is evident. It seems that there was a foramen present at the intersection of the wing-shaped to the rod-shaped part, but this could as well be a taphonomic artefact.

Figure 5 Girdle elements of Plesionectes longicollum gen. et sp. nov. (SMNS 51945).

(A) Interclavicle in ventral view. (B) Left scapula in lateral view. (C) Distorted coracoid portions. (D) Ischia in dorsal view. (E) Dorsal section of right ilium in medial view. (F) Left ilium in lateral view. (G) Fragmentary pubes in supposed ventral view. Scale bars equal three cm. Abbreviations: acf, acetabular facet; aip, anterior process of ischium; dp, dorsal process of scapula; for, foramen; gf, glenoid fossa of left coracoid; lw, lateral wing; p.co, posterior coracoid portion; p.is, posterior ischium portion; pef, pelvic fenestra; pf, pubic facet of ischium; pu, pubis; rsp, rod-shape part of interclavicular; vp, ventral plate of scapula.

Scapula. The left scapula is well-preserved and exposed in lateral view (Fig. 5B). The right scapula is obscured by the left one and only part of its dorsal process is visible. The anterior side of the ventral plate was partly repaired, but a vertical anterior edge that is almost twice as high as the glenoid end of the ventral plate seems to be genuine and matches the morphology observed in Plesiopterys wildi (see O’Keefe, 2004, fig. 6.1). The ventral side of the ventral plate is gently concave in lateral view and no lateral elaboration is formed. The posterior articular end is not well defined, possibly due to preservation, but the ventral plate becomes gently higher toward the posterior side. The dorsal process is inclined posteriorly by about 50 degrees to the horizontal plane. The anterior edge of the dorsal process is gently concave, the posterior one is straight. The dorsal side of the dorsal process bears a slightly rugose facet that is, however, not well visible. The base of the dorsal process is considerably wider than its dorsal end (see Table S3). Directly behind the left dorsal process, part of the opposite dorsal process of the right scapula is visible.

Coracoid. Only parts of the coracoids are exposed posterior to the scapulae (Fig. 5C). The elements are distorted, have been partly reconstructed and are covered by a layer of plaster. It thus remains unclear if the morphology is genuine. A large element that is placed near the dorsal process of the right scapula might be part of the distorted left coracoid, possibly the anterolateral preglenoid section, but it cannot be determined any closer. The portion of the left coracoid that is placed at the humerus indicates an elongate articular facet and might be part of the glenoid fossa (as has also been identified by Vincent et al., 2017). The remaining post-glenoid part of the left coracoid has been reconstructed. The right coracoid is placed next to the left one and the posteromedial part is exposed in dorsal view. This section is covered by plaster but it appears that the edges are genuine. Hence, the coracoids were separated posteromedially.

Ischium. Both ischia are preserved (Fig. 5D). The right ischium is partly overlain by vertebrae and the left ilium, the left ischium is partly obscured by the opposite ischium and likewise by the ilium. However, both ischia together reveal all aspects of the ischial morphology. The ischia are boot-shaped elements as in other plesiosaurs (e.g., Andrews, 1910; Andrews, 1913; Fraas, 1910; Sachs, Hornung & Kear, 2016). They have a concave anterior margin which formed the posterior frames of the pelvic fenestrae. Anteromedially, a pronounced process is formed that extends almost to the level of the lateral pubic facet. A pelvic bar was thus not fully fused. However, owing to its short distance to the pubic facet, it is likely that a fused pelvic bar would have been present in a later ontogenetic stage. The anterior edge of the anteromedial process is inclined which indicates that a rhombic fenestra was present in the middle of the pelvic bar, as seen in Brancasaurus brancai (Sachs, Hornung & Kear, 2016), Futabasaurus suzukii (Sato, Hasegawa & M, 2006), or Seeleyosaurus guilelmiimperatoris (Sachs et al., 2025). The medial symphysis is long and the posterior side is laterally inclined, which shows that the ischia were separated posteromedially. The posterior side of each ischium is stronger concave than the anterior one. Laterally, the acetabular end is exposed in the left ischium. The pubic facet appears to be smaller than the one that took part in the formation of the acetabulum.

Ilium. The left ilium is complete and exposed in lateral view next to the ischia (Fig. 5F). Of the right ilium only the dorsal end, shown in medial view, is present adjacent to the neural spines of the sacral vertebrae (Fig. 5E). The ilium has a ‘hourglass-like’ morphology with a constricted midshaft and anteroposteriorly expanded dorsal and ventral ends. Dorsally, the ilium is more widened than ventrally, but both ends are perpendicular to one another. The ventral part of the shaft is partly collapsed. The ventral end shows a rugose facet of which only part of the convex lateral side is exposed. The dorsal side of the left ilium is obscured by the sacral ribs, but the opposite ilium shows that a convex dorsal end was formed.

Pubis. Both pubes are preserved (Fig. 5G), but the elements are partly covered by repair plaster and appear incomplete. A long and straight pubic symphysis is evident. The acetabular end appears to be obscured by the left femur.

Humerus. The left humerus is preserved in dorsal view (Fig. 6A). The element has a collapsed midshaft and is considerably shorter than the femur (ratio: 0.87). At the proximal end, only the tuberosity is exposed which has a rugose surface. The preaxial margin of the humerus is gently concave but almost straight. The postaxial margin is straight for most of its length. Only distally it is slightly posteriorly curved. At the distal end, no pronounced epipodials facets are evident.

Figure 6 Limb elements of Plesionectes longicollum gen. et sp. nov. (SMNS 51945).

(A) Front limb elements. (B) Hind limb elements. (C) Close-up of the proximal front limb elements. (D) Close-up of the proximal hind limb elements. Scale bars equal three cm. Abbreviations: dc, distal carpals (I–IV); dt, distal tarsals (I–IV); fe, femur; fi, fibula; fib, fibulare; hu, humerus; int, intermedium; mc, metacarpals (I–V); mt, metatarsals (I–V); ph, phalanx; ra, radius; rad, radiale; si, spatium interosseum; ti, tibia; tib, tibiale; tro, trochanter; tub, tuberosity; ul, ulna; uln, ulnare.

Radius. Both radii are preserved; the left one is in articulation with the ulna and visible in dorsal view (Fig. 6A). The radius bears a concave preaxial margin. The postaxial margin is likewise concave and forms part of the spatium interosseum (Fig. 6C). Proximally, a gently convex edge is formed for articulation with the humerus. Distally, two slightly angled facets for the anterior articulation with the radiale and the posterior one with the intermedium are evident. The opposite radius mirrors this morphology.

Ulna. The left ulna is present in articulation with the radius (Fig. 6A). It is a crescent-shaped element that is only slightly larger than the radius. The preaxial margin is concave and participates in the spatium interosseum (Fig. 6C). Proximally, a straight edge for the humerus articulation is formed. The posterior side is convex and merges with the distal side of the element. The latter shows two angled facets, a posterior one for the ulnare and an anterior facet for the intermedium.

Carpals. The carpals are preserved in their original arrangement (Figs. 6A, 6C). All carpals are more or less circular elements, but sometimes the cortex is broken off. The ulnare is largely covered by repair plaster so the circular morphology of this element may be an artefact. The intermedium has a broad-oval appearance and the radiale is again partly damaged but seems to be circular. From the right paddle only one carpal element is evident which best resembles the intermedium in shape and size. The distal carpal I is placed distal to the radiale and the smallest element of all carpals. The distal carpal 2+3 is larger than the first one and placed distal to the intermedium. The distal carpal 4 is placed distal to the ulnare and is the largest element in the row.

Metacarpals. The metacarpals are placed in their original arrangement (Fig. 6C). The first metacarpal is very short and more rod-shaped compared with the other elements. It is placed adjacent to the second metacarpal. The latter is a robust element with a slightly concave anterior and posterior edge, a convex proximal side and a straight distal one. The second metacarpal is in line with the third metacarpal, but both seem to have moved slightly more distally to their original position. Metacarpals 3 and 4 are both robust and rectangular-shaped elements. The fifth metacarpal is placed between the distal carpal row with about half of its length. It is the largest metacarpal element and has a rectangular appearance.

Femur. Both femora are preserved (Fig. 6B). The right one is partly obscured by the right ischium and shows the ventral aspect, the left femur exposes the dorsal aspect. The femora are longer than the preserved left humerus and their distal ends are also more anteroposteriorly expanded. At the proximal end, the trochanter is exposed at the left femur which has a slightly indented and rugose surface. As in the humerus the shaft of the left femur is collapsed. The pre- and postaxial margins of the femoral shafts are straight for most of their length. The preaxial margin is only slightly anterodistally curved, whereas the postaxial one curves more prominently posterodistally. The distal articular faces are not well defined but are otherwise covered by matrix.

Tibia. Both tibiae are preserved (Figs. 6B, 6D). The left one is in the original articulation with the fibula, the right one is largely obscured by other limb elements and placed adjacent to the left tibia (Fig. 6D). The tibia has a concave preaxial margin that is longer than the postaxial one. The latter is more deeply concave and participated in the spatium interosseum. The proximal side is convex for articulation with the femur and the largest side of the element. The distal side indicates two facets for the tibiale and intermedium which are, however, not as well defined.

Fibula. The left fibula is present in articulation with the tibia, the right one is placed adjacent to the distal end of the left femur (Fig. 6D). The elements are crescent-shaped with a short and concave preaxial margin that forms part of the spatium interosseum. The proximal side is the largest of the element. It is slightly anteromedially inclined for articulation with the femur. The postaxial margin is convex and merges with the distal side. Distally, there are two barely defined facets, a posterior one for the fibulare and an anterior one for the intermedium.

Tarsals. The tarsals are disarticulated and only some of the elements can be identified (Fig. 6D). The tibiale is circular and placed adjacent to the left tibia. It is smaller than the other proximal tarsals. The intermedium is likewise placed distal to the left tibia and has a broad-oval shape. The fibulare is found next to the fibula. It has a circular shape and is the largest element of the proximal tarsal row. The first distal tarsal cannot be identified with certainty, but two circular elements placed distal to the left intermedium seem to be distal tarsal 2+3, as well as distal tarsal 4.

Metatarsals. The metatarsals 2–5 are still in the original arrangement, but metatarsal 1 cannot be defined with certainty (Fig. 6D). The metatarsals are robust elements but they are shorter than the metacarpals. They all have a slightly concave anterior and posterior edge. Metatarsals 2–4 have gently convex proximal and distal sides and metatarsal 2 is slightly smaller than metatarsals 3–5. The fifth metatarsal is placed proximally with half of its length. The proximal edge is slightly anterodistally inclined for the articulation with the intermedium.

Phalanges. The phalanges are present in the original arrangement in the left front and hind limb (Figs. 6A–6D). They are disarticulated in the corresponding right limbs. In both limbs, no first digit is formed. The second digit bears six phalanges in the left front and hind limb. The third digit has eight phalanges in the left front limb and 10 in the corresponding hind limb. The fourth digit bears eight phalanges in the left front limb and nine in the left hind limb, and digit five has seven phalanges in the left front limb and eight in the left hind limb. All phalanges are robust and hourglass-shaped with concave anterior and posterior margins. They are not considerably proximodistally expanded. The distalmost phalanges are conical in shape.

Results of phylogenetic analyses

Numerical results of the phylogenetic analyses and full tree topologies are provided in Table 2 and Supplemental Information 3, respectively. All three analyses support the placement of Plesionectes longicollum gen. et sp. nov. among early-diverging plesiosauroids though its position differs across particular runs (Fig. 7). Equal weighting resulted in a poorer resolution of the strict consensus tree. However, a closer inspection of the 50% majority-rule consensus tree under equal weighting revealed that 100% of the most parsimonious trees include the new taxon within the least inclusive node comprising Microcleididae and later-diverging plesiosauroids, with its preferred position as the sister taxon to the clade formed by Franconiasaurus brevispinus and Cryptoclidia (Fig. 7A). In contrast, both analyses using the implied weighting function (K = 9 and 12) found P. longicollum as part of a clade formed by Plesiopharos moelensis and Stratesaurus taylori (Fig. 7B), which was situated outside the smallest clade consisting microcleidids and later-diverging forms.

Table 2 Numerical results of the parsimony analyses.

BS, best score (tree length); CI, Consistency Index; EW, parsimony analysis using equal weighting; IW, parsimony analysis using implied weighting; MPT, number of most parsimonious trees; NT, ‘New Technology’ search; RI, Retention Index; TS, ‘Traditional’ search.

Run	MPT (NT)	BS	MPT (TS)	CI	RI	
EW	55	2,120	200,000	0.188	0.685	
IW ( K = 9)	17	110.56244	96,957	0.187	0.682	
IW ( K = 12)	12	92.16791	161,595	0.187	0.683	

Figure 7 The phylogenetic placement of Plesionectes longicollum gen. et sp. nov. (SMNS 51945) showed on the plesiosauroid branch of Plesiosauria.

(A) Reduced strict consensus tree reconstructed through parsimony analysis using equal weighting (numbers at nodes indicate the Bremer support values) and (B) reduced strict consensus tree reconstructed through weighted parsimony analyses with K set to 9 and 12. The arrow indicates the extent of Microcleididae. The full tree topologies inferred in particular parsimony analyses and the results of Symmetric Resampling are provided in Supplemental Information 3.

Discussion

Ontogenetic stage of SMNS 51945

SMNS 51945 shows several characters that have been reported in osteologically immature (sensu Araújo et al., 2015, Araújo & Smith, 2023) plesiosaurs. These include the unfused neurapophysis throughout the vertebral column, the barely developed epipodial facets at the propodials (sensu Brown, 1981), and the collapsed propodial shafts which indicates an incomplete ossification. However, the pronounced anteromedial process at the ischium, potentially leading to the development of a pelvic bar, is a character that is typically considered to be found in later osteological stages (sensu Brown, 1981). Although ‘osteological (im)maturity’, as used by Araújo et al. (2015) and Araújo & Smith (2023), should not be associated with certain ontogenetic stages, we consider the character state distribution in SMNS 51945 to indicate that the individual was not a young juvenile, which is further supported but the large size of the specimen. The skeleton, as preserved, measures 2.95 m and the complete skeleton had likely measured about 3.20 m in length (Großmann, 2006, appendix B, even considered a complete length of 3.55 m). The contemporary plesiosauroid Seeleyosaurus guilelmiimperatoris measures between 2.88 m (MB.R.1992) and 3.40 m (SMNS 12039) (Dames, 1895; Fraas, 1910; Sachs et al., 2025), Microcleidus brachypterygius (GPIT-PV-60640) measures 2.99 m (von Huene, 1923), and Plesiopterys wildi measure between 2.20 m (SMNS 16812) and 3.00 m (MH 7) (O’Keefe, 2004; Marx et al., 2025).

Diagnostic characters of Plesionectes longicollum gen. et sp. nov.

We are inclined to agree with Araújo et al. (2015) that the terms ‘juvenile’ and ‘adult’ should preferably not be used for plesiosaurs without osteohistological analysis. Nonetheless, there are still cases in which it is possible to determine whether one individual is older or younger than another representative of the same taxon. The 2.20-meter-long holotype of Plesiopterys wildi (SMNS 16812) is a much younger individual than the 3.00-meter-long referred specimen (MH 7) of that taxon (Marx et al., 2025). While some of the differences between these two individuals certainly represent intraspecific variation (MH 7 is also from slightly younger strata), ontogeny undoubtedly impacts their character state distribution as well. Owing to the fact that the identification of potential ontogeny-dependent differences are essential especially from the taxonomic and phylogenetic viewpoint, we briefly assessed the variation within Plesiopterys within the framework of the character matrix of Benson & Druckenmiller (2014) that we used to infer the phylogenetic placement of Plesionectes gen. nov. From the 270 characters assembled by Benson & Druckenmiller (2014), 112 are assessable in both Plesiopterys specimens (SMNS 16812 and MH 7). Of these, 16 (∼14%) show some degree of variation which leads to different character scores. The majority of these differences (11 characters) are found in the girdle elements and limbs. The remaining scores (∼86%) are identical.

It is worth noting that the type and referred specimens of Seeleyosaurus guilelmiimperatoris and Franconiasaurus brevispinus also exhibit size differences, with the holotypes in both taxa being smaller than the referred specimens (Sachs, Eggmaier & Madzia, 2024; Sachs et al., 2025). In Seeleyosaurus guilelmiimperatoris, only four (∼3%) of 122 characters assessable in both specimens (MB.R.1992 and SMNS 12039) differ. In Franconiasaurus brevispinus, 58 characters are assessable in both the type (BT 011224.00) and referred specimen (BT 011241.00), and only three of them (∼5%) differ.

The characters, which we consider diagnostic for Plesionectes longicollum gen. et sp. nov., are identical in the specimens of P. wildi, S. guilelmiimperatoris, and F. brevispinus. This indicates that, while some minor variation may occur, their overall appearance is likely stable during ontogeny.

Below we discuss each of these characters separately.

Paraoccipital process being considerably longer as the height of the exoccipital body. The paraoccipital process in SMNS 51945 is considerably longer than the height of the exoccipital body (1.5 times as preserved). A similarly long paraoccipital process that is at least 1.3 times as long as exoccipital height is found, e.g., in the Toarcian microcleidid Microcleidus homalospondylus (Brown, Vincent & Bardet, 2013, fig. 3.2) or some later-diverging plesiosauroids, such as Tricleidus seeleyi (Brown, 1981, fig. 23) and Umoonasaurus demoscyllus (Kear, Schroeder & Lee, 2006, fig. 1d). A shorter paraoccipital process that is subequal to the height of the exoccipital body is present in Plesiosaurus dolichodeirus (Benson & Druckenmiller 2024, Appendix 2, character 70) and Plesiopterys wildi (O’Keefe, 2004, fig. 5, Marx et al., 2025, fig. 5).

Neck comprising more than 40 cervicals. The neck of SMNS 51945 is articulated and includes 41 post-axial vertebrae. The complete neck thus comprised at least 43 vertebrae in which it remains unclear if only the atlas-axis or one or two more anteriormost cervical vertebrae are missing. However, even the minimum count of 43 cervicals is higher than the cervical count in most other Early Jurassic plesiosauroids with less than 30 in Franconiasaurus brevispinus (Sachs, Eggmaier & Madzia, 2024), 35–36 in Seeleyosaurus guilelmiimperatoris (Sachs et al., 2025), 36 in Eoplesiosaurus antiquior (Benson, Evans & Druckenmiller, 2012) and Microcleidus brachypterygius (Von Huene, 1923), 39 in Plesiopterys wildi (O’Keefe, 2004), 40 in Microcleidus homalospondylus (Watson, 1909), and 41 in Plesiosaurus dolichodeirus (Storrs, 1999). Only Microcleidus tournemirensis has an equally or similarly high number with 43 cervical vertebrae (Bardet, Godefroit & Sciau, 1999). A similar number with 44 cervicals was described for the later-diverging plesiosauroid Muraenosaurus leedsii (Brown, 1981).

V-shaped neurocentral suture developed in the cervical and pectoral vertebrae. The neurocentral suture has a V-shape in all cervicals, but also in the pectoral vertebrae. A corresponding suture has been described for anterior and mid-cervicals of the early plesiosaur Rhaeticosaurus mertensi (Wintrich et al., 2017), and it is known in rhomaleosaurids, such as Thaumatodracon wiedenrothi (Smith & Araújo, 2017), Eurycleidus arcuatus and Rhomaleosaurus cramptoni (Benson & Druckenmiller, 2014, Appendix 2, character 172). A V-shaped neurocentral suture is also present in the posterior cervical and pectoral vertebrae of Plesiopharos moelensis (Puértolas-Pascual et al., 2021). In the contemporary plesiosauroids Microcleidus tournemirensis, Microcleidus brachypterygius, Microcleidus homalospondylus, and Plesiopterys wildi the suture is rounded (Benson & Druckenmiller, 2014, Appendix 2, character 172). In Plesiosaurus dolichodeirus, Eoplesiosaurus antiquior, and Westphaliasaurus simonsensii (Benson & Druckenmiller, 2014, Appendix 2, character 172) the neurocentral suture is evenly convex and extends further ventrally than in SMNS 51945. The neurocentral suture is well ossified in Franconiasaurus brevispinus (Sachs, Eggmaier & Madzia, 2024) and in the type specimen of Seeleyosaurus guilelmiimperatoris (Sachs et al., 2025).

Combination of two conjoint rib facets in anterior, mid, and the majority of the posterior cervicals, and one rib facet in the posteriormost cervical vertebrae. A combination of cervical rib facets with a conjoint parapophysis and diapophysis in most of the cervical vertebrae, but one rib facet in the posteriormost cervicals is, to our knowledge, only known in Westphaliasaurus simonsensii (S. Sachs, pers. obs., 2024) among plesiosauroids. A similar condition has been described for some Middle and Late Jurassic pliosaurids including Marmornectes candrewi, Peloneustes philarchus, Eardasaurus powelli, and Pliosaurus brachyspondylus (Benson & Druckenmiller, 2014, Appendix 2, characters 160 and 161, Ketchum & Benson, 2022). The opposite condition with a single cervical rib facet in the anterior cervicals and two conjoined facts in the posterior cervicals was reported for the microcleidid Microcleidus melusinae (Vincent et al., 2019) and the rhomaleosaurid Lindwurmia thiuda (Vincent & Storrs, 2019).

Pronounced cervical rib processes in anterior and mid-neck region. SMNS 51945 bear pronounced anterior and posterior cervical rib processes in the anterior and mid-neck cervicals. This condition is shared with microcleidids (Benson & Druckenmiller, 2014, Appendix 2, character 163), but distinguishes the specimen from Westphaliasaurus simonsensii (Schwermann & Sander 2011), Franconiasaurus brevispinus (Sachs, Eggmaier & Madzia, 2024), and Plesiopterys wildi (Marx et al., 2025) where the anterior cervical processes are reduced.

Posterior cervical and pectoral neural spines not considerably taller than long and lacking constriction at base. This character distinguishes Plesionectes longicollum gen et sp. nov. from contemporary microcleidids where the neural spines are considerably taller than long and have a constricted base (Benson, Evans & Druckenmiller, 2012). The neural spines in Plesionectes longicollum gen. et sp. nov. resemble those described for Plesiopharos moelensis, Westphaliasaurus simonsensii and Franconiasaurus brevispinus (Puértolas-Pascual et al. 2021, Schwermann & Sander 2011; Sachs, Eggmaier & Madzia, 2024). Neural spines are known to vary ontogenetically (Brown, 1981) and it is possible that they would have been taller in a later ontogenetic stage, but it is unlikely that they would have reached an extreme height/length ratio as in Seeleyosaurus guilelmiimperatoris (Sachs et al., 2025) or Microcleidus tournemirensis (Bardet, Godefroit & Sciau, 1999) where they are up three times as tall as long.

High number of dorsal vertebrae. SMNS 51945 bears 20 to 21 dorsal vertebrae. This number is higher than in most other contemporary plesiosauroids. The dorsal series is composed of 14 vertebrae in Microcleidus brachypterygius (GPIT-PV-60640, (Großmann, 2007). Seeleyosaurus guilelmiimperatoris bears 15 (in SMNS 12039) and 17 (in MB.R.1992) dorsals (Sachs et al., 2025). For Microcleidus tournemirensis 16 dorsal vertebrae have been described (Bardet, Godefroit & Sciau, 1999), and 17 for Microcleidus homaleospondylus (Watson, 1909). An equally high number is, however, present in Plesiopterys wildi with 19 dorsals (O’Keefe, 2004), and Franconiasaurus brevispinus with 22 dorsal vertebrae (Sachs, Eggmaier & Madzia, 2024).

Additional potentially diagnostic characters

Additional potentially diagnostic characters include the transverse processes of the dorsal vertebrae which are very short when compared with the contemporary taxa Microcleidus tournemirensis (Bardet, Godefroit & Sciau, 1999), Plesiopterys wildi (S. Sachs, pers. obs., 2024), and Seeleyosaurus guilelmiimperatoris (Sachs et al., 2025). Comparably short transverse processes are, however, found in M. melusinae (S. Sachs, pers. obs., 2024). The interclavicle has a peculiar shape too in which the anteromedial side is deeply notched instead of widely concave as in Seeleyosaurus guilelmiimperatoris (Sachs et al., 2025), Westphaliasaurus simonsensii (Schwermann & Sander 2011) or Microcleidus tournemirensis (Bardet, Godefroit & Sciau, 1999). It is, however, unclear in what way these conditions would have changed in a later ontogenetic stage.

Figure 8 Schematic stratigraphic distribution of nearly complete plesiosaur skeletons in the lower Toarcian (upper Lower Jurassic) Lias ɛ II sections of the Posidonia Shale (Posidonienschiefer Formation) in the Holzmaden area, Southwest Germany.

Silhouette obtained from phylopic.org (Adam Stuart Smith, vectorized by T. Michael Keesey; CC BY-SA 3.0).

Stratigraphic distribution of plesiosaur specimens from the Holzmaden area

The plesiosaur specimens from the lower Toarcian of the Posidonienschiefer Formation cropping out in the Holzmaden area were unearthed from strata spanning nine Lias ɛII sections (Lias ɛII1−9) belonging to three subzones of the Dactylioceras tenuicostatum and Harpoceras falciferum ammonite zones (D. semicelatum of the D. tenuicostatum zone and H. exaratum and H. falciferum of the H. falciferum zone) (Hauff, 1921; Riegraf, Werner & Lörcher, 1984; Maisch, 2021; Fig. 8). The majority of Holzmaden plesiosaurs have been unearthed from Lias ɛII4 within the exaratum subzone (elegantulum horizon). These include, among others, the types of Hauffiosaurus zanoni (MH uncataloged), Microcleidus brachypterygius (GPIT-PV-60640), and Plesiopterys wildi (SMNS 16812). The type and referred specimens of Seeleyosaurus guilelmiimperatoris (MB.R.1992 and SMNS 12039, respectively) are slightly older, and originate from Lias ɛII3 within the antiquum horizon of the semicelatum subzone, while the holotype of Meyerasaurus victor (SMNS 12478) and referred specimen of P. wildi (MH 7) derive from Lias ɛII5 and Lias ɛII6, respectively, within the exaratum subzone and are therefore the youngest of the diagnosable plesiosaur specimens described so far. The description of SMNS 51945 (Plesionectes longicollum gen. et sp. nov.) from Lias ɛII1 within the semicelatum horizon of the semicelatum subzone further extends the stratigraphic range of the Holzmaden area plesiosaurs, reporting the oldest known Holzmaden taxon.

Concluding remarks

The lower Toarcian (upper Lower Jurassic) Posidonia Shale at Holzmaden, located in Southwest Germany, has produced some of the most remarkable specimens of marine tetrapods, including representatives of all three major plesiosaur lineages: the plesiosauroids (Microcleidus brachypterygius, Plesiopterys wildi, Seeleyosaurus guilelmiimperatoris), a pliosaurid (Hauffiosaurus zanoni), and a rhomaleosaurid (Meyerasaurus victor), each known from nearly complete skeletons. Our assessment of an osteologically immature plesiosauroid skeleton, discovered in a Holzmaden quarry in 1978, housed at the collections of Staatliches Museum für Naturkunde Stuttgart, Stuttgart, Germany, and cataloged under the number SMNS 51945, revealed that the specimen displays an unusual combination of character states (including one potential local autapomorphy), which do not appear to be significantly influenced by ontogenetic development. This distinctive set of features warrants the establishment of a new taxon, which we have named here Plesionectes longicollum gen. et sp. nov.

The new taxon originates from Lias ɛII1 (tenuicostatum zone; semicelatum subzone; lower Posidonienschiefer Formation) and is therefore the oldest known plesiosaur specimen from the Holzmaden area, enhancing our understanding of plesiosaur diversity during the Toarcian age but also contributing to the broader knowledge of biodiversity present in the Posidonia Shale, which still remains incompletely known.

Supplemental Information

Supplemental Information 1 Character list for the phylogenetic analyses of Plesiosauria

Supplemental Information 2 Character matrix for the phylogenetic analyses of Plesiosauria

TNT-executable script.

Supplemental Information 3 Full results of the parsimony analyses

Supplemental Information 4 Measurements (in mm) of the axial skeleton of Plesionectes longicollum (SMNS 51945)

Supplemental Information 5 Measurements (in mm) of the limb elements of Plesionectes longicollum (SMNS 51945); dist = distally, prox. = proximally

Supplemental Information 6 Measurements (in mm) of the girdle elements of Plesionectes longicollum (SMNS 51945)

We would like to express our gratitude to Erin Maxwell for providing access to the specimens under her care, Günter Schweigert (both SMNS) for discussion on Holzmaden stratigraphy, and Katrin Sachs (Engelskirchen, Germany) for her tireless and irreplaceable assistance during the measuring of the specimen. We are further indebted to Academic Editor Dagmara .Zyła (Museum of Nature Hamburg, Leibniz Institute for the Analysis of Biodiversity Change, Hamburg, Germany) for handling our manuscript and two anonymous reviewers for their constructive comments that helped us to improve the manuscript. TNT is made freely available with the sponsorship of the Willi Hennig Society.

Institutional abbreviations

BT Urwelt-Museum Oberfranken, Bayreuth, Germany

GPIT Geologisch-Paläontologisches Institut, Eberhard Karls Universität Tübingen, Tübingen, Germany

MB Museum für Naturkunde Berlin, Berlin, Germany

MH Urwelt-Museum Hauff, Holzmaden, Germany

SMNS Staatliches Museum für Naturkunde Stuttgart, Stuttgart, Germany

Additional Information and Declarations

Competing Interests

Author Contributions

Data Availability

New Species Registration

The authors declare there are no competing interests.

Sven Sachs conceived and designed the experiments, performed the experiments, analyzed the data, prepared figures and/or tables, authored or reviewed drafts of the article, and approved the final draft.

Daniel Madzia conceived and designed the experiments, performed the experiments, analyzed the data, prepared figures and/or tables, authored or reviewed drafts of the article, and approved the final draft.

The following information was supplied regarding data availability:

The character matrix necessary to replicate the phylogenetic analyses is available in the Supplementary File.

The following information was supplied regarding the registration of a newly described species:

Publication LSID: urn:lsid:zoobank.org:pub:23DBDB92-0752-4068-ACC5-A29D217 D6CE3

Plesionectes longicollum species name: urn:lsid:zoobank.org:act:73D43879-D9DA-410A-8F2D-F48BC1033432.

Plesionectes genus name: urn:lsid:zoobank.org:act:C1A1E982-0895-46F5-9B21-F9741E111949.

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
