# Peer review of "An unusual early-diverging plesiosauroid from the Lower Jurassic Posidonia Shale of Holzmaden, Germany"

_PeerJ, doi:10.7717/peerj.19665_

## Round 0.1 · original submission · Minor Revisions

Please, take all the reviewers' comments into account. Pay attention to the taxonomic doubts raised by one of the reviewers.

Reviewer 1 ·

Basic reporting

The English is overall clear, although some word choice is uncommon or confusing. For example, “post-axis” instead of “postaxial”; “gastric ribs” instead of gastralia; “linked” for fused/connected structures; “mid-section of the tail” likely mid-length?; “prezygapophyses appear to exceed the according centrum” and “postzygapophyses are only exceeding the centra with about half of their length” – unclear, likely implies zygapophyses protruding far anteriorly than centrum articular face?; “hour-clock” – hourglass?; frequent use of “next to”, it seems that sometimes it can be replaced by “near”, “close to” etc.; frequent use of “likewise” – consider rephrasing and synonyms; “ventral shaft”???; “Only distally there it is slightly posteriorly curved” - ???; “tuberosity is visible which merges with the shaft” – how so tuberosity merges with the shaft?; use of “according” e.g. “according centrum”, “according limb” ???; “biodiversity present in the Posidonia Shale, which still remains incompletely recognized” – see “recognize” in a dictionary.
Sufficient background and context are provided. Cited literature is relevant. Among the small issues is that authors not always cite works in which latest taxonomic decisions were made, e.g. Seeleyosaurus guilelmiimperatoris (Dames 1895, Fraas 1910) and Microcleidus brachypterygius (von Huene 1923, Großmann 2006, 2007) - White (1940) and Großmann (2007) should be cited for taxonomic purposes, as in cited papers S. guilelmiimperatoris is referred to other genera; same for M. brachypterygius – cite Benson et al. (2012).
Article structure, figures and tables match common standards. In particular, the strength of this contribution is detailed photographs of the specimen and all described structures. Raw data are shared.
All appropriate raw data have been made available in accordance with our Data Sharing policy.
The contribution is self-contained and includes all results relevant to the hypothesis.

Experimental design

This is original primary research within aims and scope of the journal. It is relevant and meaningful contribution conducted in conformity with the prevailing technical and ethical standards in the field. Methods are described with sufficient information to be reproducible by another investigator.

Validity of the findings

Authors clearly explain why they consider the specimen a new taxon and discuss their hypothesis. I agree that the specimen represents a new genus and species, and that the taxon can be erected despite the osteological immaturity of the holotype.

Additional comments

Additional minor remarks
Line 33 – from the Posidonia shale
Line 109 - although it is authors' decision how to name the taxon, the proposed name is etymologically confusing. How to interpret near swimmer? That its swimming capabilities were not that good as in other plesiosaurs?
Line 118 “partial skull” – partially exposed skull - the skull may be complete, but not exposed, obscured by the neck and sediment
Line 129 etc, Diagnosis, - why not to add the reduced first digit in fore- and hindlimbs – it looks very interesting feature. Also recommend to discuss this in respective section.
Line 152 “the dorsally placed element, identified as part of the mandible, actually represents the squamosal arch” - please, be less confident in your phrasing. To reader, your "actual" interpretation looks not more convincing than that of Vincent et al. (2017), even though yours seems more likely.
See attached my speculative interpretation based on your photo. There may be not just squamosals, but also articulated quadrates in glenoids and exposed retroarticular processes of the mandible…
Lines 163-164 – agree that quadrates seem to be present
Lines 173 and 179. What you describe is basic structure and a series of hypoglossal foramina (XII) at the base of the exoccipital peduncle are common not just for plesiosaurs, but present like that in all amniotes. These are described in many papers, thus it is strange to cite only Benson et al. 2015 (e.g. Benson et al 2011 also depict exoccipital-opisthotic in detail for Thalassiodracon; there are papers by other authors providing interpretations of these foramina, like Carpenter 1997).
Which one of the two papers of Sachs et al (2016) is cited. Add a and b, or cite three authors, as common for PeerJ. It is always easy to remember your own papers and cite them, but there are many papers proposing interpretations of foramina in the exoccipital-opisthotic. This is odd to cite only Sachs et al 2016 here. They are not the people who first interpreted this in plesiosaurs, neither those who did this with any anatomical substantiation.
Lines 186-187. You have no serious reasons to conclude that several cervical vertebrae are missing. You can't even say that the axis is missing, because in osteologically immature specimens, the morphology of the axis is like that of the third centrum, particularly if the axial intercentrum is poorly developed. See e.g. Andrews 1910, fig 78. Thus, I agree with Vincent et al 2017 and count 42-43 cervical vertebrae.
Lines 200 - 201. “we are confident that a lateral ridge was indeed preserved and is found at about the same position in all of the centra” - These are speculations, worth nothing in the description, even if your guess is correct. CT or preparation from the opposite side is needed to make statements like this.
Lines 201-202. Brown studied only a limited number of Callovian taxa primarily. His observatios should be extrapolated with caution.
Line 204 – osteologically immature. Throughout I recommend following Araújo (et al., 2015; - & Smith, 2023) and refer to osteological (im)maturity.
Lines 204-205 “a lateral longitudinal ridge it was described for” – delete “it”
Lines 212 - 216. How so you first say that the base is not narrower than the distal end, citing Benson et al, and right after that report distal expansion of the distal ends? Don't you think it looks self-contradictory?
Line 234. “enigmatic” – redundant
Line 239. “bear a distinct anterior and posterior process” - bear distinct anterior and posterior processes.
Line 241. “In the posterior region” - In the posterior cervical region
Line 304. I prefer “gastralia” rather than common name
Lines 317-318. which count? You are uncertain above. Just be consistent - e.g. "Thus, the exact count is unclear, but count reported by Vincent et al. (2017) seems less likely....
Line 341. “provided by Großmann” – reported?
Line 391-392 “becomes gently higher toward to the posterior side.” “to” is extra here
Line 433. “hour-clock like morphology” ???

Line 435. “Dorsally the ilium is more widened than the ventral side” – replace by “ventrally” for consistency
Line 435. what is ventral shaft?
Line 436. “rugose convex facet” – “rugose convex surface”? As it is convex, there may be two poorly demarcated facets?
Line 448 “Only distally there it is slightly posteriorly curved” something is extra here… maybe “there”?
Line 449 tuberosity merges with the shaft - how so?
Lines 456-457. “The opposite radius mirrors this morphology” – redundant
Line 459. “The left ulna is present in articulation with the radius” – redundant, mentioned above
Lines 563-564. “In conclusion, the preserved skeletal features indicate that SMNS 51945 was not a mature individual, but rather a late juvenile or early subadult.” and line 567. “ontogenetically immature (sensu Araújo & Smith 2023)” - re-read Araújo et al. 2015 and Araújo & Smith 2023. Do not confuse ontogenetic state and osteological maturity
Sections below. “several characters that are known in plesiosaurs to be ontogenetically stable” - provide relevant references that the character you list below are “ontogenetically stable”. For example, exoccipital process length - actually it gives nothing to ontogeny. not discussed how it is related to ontogeny to be stable or not. No evidence or relevant reference provided that it is shorter in younger individuals.
Lateral keel - You report a variation of this feature in Seeleyosaurus, like this is not a problem, and not discuss it. And then infer from a few exposed incipient keels on the specimen that in the case of SMNS 51945 lateral keels are taxonomically important…
A side note. “Sachs et al. in review” is cited too often, it is problematic for peer review. Hopefully, it will be published and replaced by relevant reference. Otherwise, I’d recommend to refer to pers. obs. for all these instances.
Lines 620-621. neurocentral suture <…> is also well ossified??? You mean closed? Elements co-ossified with no suture visible?
Line 665. “Middle Jurassic cryptocleidids” – you cited papers on Late Jurassic cryptoclidids.
Figure 2 caption. “Preserved skull elements” – do not confuse preserved and exposed – there may be a complete skull inside of rock.
Fig. 3 fs – fs - “foramen subcentrale” – subcentral foramen?

Annotated reviews are not available for download in order to protect the identity of reviewers who chose to remain anonymous.

Reviewer 2 ·

Basic reporting

The manuscript is generally well written, and the language used is clear and professional. The anatomical descriptions are detailed and presented with sufficient clarity. However, there are a few areas for improvement:

The use of terminology such as "cervicals" to refer to cervical vertebrae is incorrect. These terms refer to body regions, not bones. The terms "vertebrae," "ribs," or "neural spines" should follow anatomical region terms for accuracy.
In the stratigraphic distribution section, it would be beneficial to provide an estimated age in millions of years (Ma) and include broader comparisons across regions and time periods (e.g., Plesiopharos, Franconiasaurus, Westphaliasaurus, Microcleidus tournemirensis, Microcleidus homalospondylus).
The conclusion section currently reads more like an abstract than a summary of the study’s findings. The focus should be on new data and conclusive results specific to this study.
The photos in Figure 2 lack clarity in showing bone boundaries. Adding interpretive drawings or dashed lines to highlight these features would significantly improve the figure.
Literature references are adequate, though comparisons with similar taxa could be further expanded in the differential diagnosis section.
Overall, the article provides sufficient background and context, although broader taxonomic comparisons are suggested. The article structure is appropriate, and data are adequately shared.

Experimental design

The research falls within the scope of the journal and presents original primary research. However, there are issues with the justification for naming a new taxon:

The main concern is the lack of clear diagnostic characters that would justify the erection of a new genus and species. All proposed diagnostic characters are shared with other plesiosaurs, and no unambiguous autapomorphy is presented. If a new taxon is to be named, the authors must identify unique features specific to this specimen.
The manuscript should include a more rigorous differential diagnosis, especially comparing the specimen with Plesiopharos (Puértolas-Pascoal et al., 2022), Franconiasaurus, and Westphaliasaurus.
The character “neural spines not considerably elongate” requires clarification—whether the elongation is dorsoventral or axial—and should be numerically quantified using measurements and proportions. The current description is insufficient to establish its relevance.
In the methods section, providing GPS coordinates for the type locality would be a useful addition.

Validity of the findings

The manuscript presents important new data, but the evidence supporting the proposal of a new taxon is insufficient as it stands:

No unique diagnostic characters (autapomorphies) are presented that distinguish this specimen from other plesiosaurs. The authors need to find characters exclusive to the taxon to substantiate their claim.
Phylogenetic position alone is not enough to justify naming a new genus, especially given the instability of different analyses shown in Figure 7.
The conclusion section could be more focused on the unique contributions of the study and should refrain from reiterating general knowledge already available from previous research.
The conclusions do not fully align with the original research question, as the support for naming the new taxon is not adequately established. More robust data are required to validate the authors’ claims.

Additional comments

Lines 573-579: The character mentioned is also shared with Muraenosaurus.
Lines 581-591: Muraenosaurus has 44 cervical vertebrae, which should be noted in the comparisons.
Line 915: Reword "ventral squamosal" to "squamosal in ventral view."
Line 948-949: Include an indication that this is a dorsal view.
Lines 925 and 927: Use the term "neural spine" instead of "posterior cervical neural spine" and "anterior cervical neural spine" for consistency.
In sum, while the article presents significant descriptive data, the justification for the new taxon needs stronger evidence in the form of unique characters and more comprehensive comparative diagnoses. The conclusions could also be condensed to reflect only the novel findings of this study.

---

## Round 0.2 · Minor Revisions

Please, look at the minor comments and suggestions of the reviewer and implement them.

Reviewer 2 ·

Basic reporting

The manuscript is clearly written and professionally presented, with appropriate use of English and a logical structure conforming to PeerJ’s standards. The background provides adequate context, and the references are current and relevant. Figures are generally informative and well labeled, and the raw data (including the TNT matrix and character list) is made available in the supplementary materials.

Nevertheless, There are a few things I recommend your attention for improvment:
The diagnosis section (lines 134–142) should explicitly state whether the listed features are included in the phylogenetic character matrix and, if so, reference character numbers.
A summary table comparing the new taxon to other Toarcian plesiosauroids (e.g., Microcleidus, Franconiasaurus, Plesiopterys, Plesiopharos, Westphaliasaurus) is recommended to highlight diagnostic differences.

Experimental design

The phylogenetic analysis is methodologically sound, using a well-established matrix and multiple parsimony approaches (equal and implied weights). The analysis is reproducible with the supplied files and includes appropriate support values (Bremer support and GC frequencies).

BUT:
The justification for favoring the implied weighting topology (which places Plesionectes outside Microcleididae) over the equal weights topology (where it nests within microcleidids) is underdeveloped. The authors should better explain why they consider the implied weights result more reliable or significant.

It would strengthen the manuscript to explicitly test whether the new characters proposed for diagnosis have an effect on topology (e.g., via constraint or pruning tests).

Although the immature condition of the specimen is acknowledged, a more detailed discussion of ontogenetic variation—particularly in features like vertebral spine proportions—is necessary to support taxonomic conclusions.

Validity of the findings

The manuscript offers a thorough anatomical description of a nearly complete Toarcian plesiosaur, SMNS 51945, and suggests it represents a new genus and species based on a combination of features. The authors are transparent about the limitations of preservation and previous studies.
The specimen in spectacular and certainlly worth being properlly described and ascribed to the right taxonomy (either new or not).

However, several concerns limit the strength of the taxonomic conclusion:

No unambiguous autapomorphy is identified for Plesionectes longicollum. All diagnostic features listed are either shared with other taxa or represent plesiomorphic conditions.

The cervical vertebral count (43), while high, is not exceptional compared to known taxa such as Muraenosaurus or Plesiopterys, and does not alone justify the specific epithet or taxonomic distinctiveness.

The “V-shaped neurocentral suture”, noted as a potential autapomorphy, is not unique and is reportedly shared with other specimens (e.g., Franconiasaurus).

Without clearer diagnostic justification, the taxonomic act of naming a new genus and species may be premature. It might be more appropriate to refer the specimen to cf. Microcleidus or as an indeterminate microcleidid until more material or clearer autapomorphies are identified.

I don't think this is Plesiopharos (quite different vertebrae, for example) but it is its sister taxon (in one of the cladograms) and a better comparison is advised.

Additional comments

Figure 2 would benefit from interpretive outlines or overlays to help clarify bone boundaries and tentative identifications.
Quantify claims such as “paraoccipital process considerably longer than exoccipital body height” with actual measurements or ratios, and note whether these traits were scored in the matrix.
The manuscript would benefit from a concise table summarizing key anatomical differences between Plesionectes and similar Early Jurassic plesiosaurs.
The unique soft-tissue preservation, although not central to this study, should at least be acknowledged briefly for completeness.

---

## Round 0.3 · accepted · Accept

Thank you for addressing the reviewer's comments. I am happy with the current version of the manuscript and recommend it for publication.